# Polyvalent Mannuronic Acid-Coated Gold Nanoparticles for Probing Multivalent Lectin–Glycan Interaction and Blocking Virus Infection

**DOI:** 10.3390/v17081066

**Published:** 2025-07-30

**Authors:** Rahman Basaran, Darshita Budhadev, Eleni Dimitriou, Hannah S. Wootton, Gavin J. Miller, Amy Kempf, Inga Nehlmeier, Stefan Pöhlmann, Yuan Guo, Dejian Zhou

**Affiliations:** 1School of Chemistry and Astbury Centre for Structural Molecular Biology, University of Leeds, Leeds LS2 9JT, UK; rahman.basaran@gmail.com (R.B.); dbudhadev@gmail.com (D.B.); 2Department of Biology, Faculty of Science, Ankara University, Ankara 06100, Türkiye; 3Centre for Glycoscience and School of Chemical and Physical Sciences, Keele University, Keele ST5 5BG, UK; elenid892@gmail.com (E.D.); h.s.wootton1@keele.ac.uk (H.S.W.); g.j.miller@keele.ac.uk (G.J.M.); 4Infection Biology Unit, German Primate Center–Leibniz Institute for Primate Research, 37077 Göttingen, Germany; akempf@dpz.eu (A.K.); inehlmeier@dpz.eu (I.N.); spoehlmann@dpz.eu (S.P.); 5Faculty of Biology and Psychology, Georg-August-University Göttingen, 37073 Göttingen, Germany; 6School of Food Science and Nutrition and Astbury Centre for Structural Molecular Biology, University of Leeds, Leeds LS2 9JT, UK

**Keywords:** mannuronic acid, Ebola virus inhibition, fluorescence quenching, gold nanoparticle, multivalent lectin–glycan interaction

## Abstract

Multivalent lectin–glycan interactions (MLGIs) are vital for viral infection, cell-cell communication and regulation of immune responses. Their structural and biophysical data are thus important, not only for providing insights into their underlying mechanisms but also for designing potent glycoconjugate therapeutics against target MLGIs. However, such information remains to be limited for some important MLGIs, significantly restricting the research progress. We have recently demonstrated that functional nanoparticles, including ∼4 nm quantum dots and varying sized gold nanoparticles (GNPs), densely glycosylated with various natural mono- and oligo- saccharides, are powerful biophysical probes for MLGIs. Using two important viral receptors, DC-SIGN and DC-SIGNR (together denoted as DC-SIGN/R hereafter), as model multimeric lectins, we have shown that α-mannose and α-manno-α-1,2-biose (abbreviated as Man and DiMan, respectively) coated GNPs not only can provide sensitive measurement of MLGI affinities but also reveal critical structural information (e.g., binding site orientation and mode) which are important for MLGI targeting. In this study, we produced mannuronic acid (ManA) coated GNPs (GNP-ManA) of two different sizes to probe the effect of glycan modification on their MLGI affinity and antiviral property. Using our recently developed GNP fluorescence quenching assay, we find that GNP-ManA binds effectively to both DC-SIGN/R and increasing the size of GNP significantly enhances their MLGI affinity. Consistent with this, increasing the GNP size also significantly enhances their ability to block DC-SIGN/R-augmented virus entry into host cells. Particularly, ManA coated 13 nm GNP potently block Ebola virus glycoprotein-driven entry into DC-SIGN/R-expressing cells with sub-nM levels of *EC*_50_. Our findings suggest that GNP-ManA probes can act as a useful tool to quantify the characteristics of MLGIs, where increasing the GNP scaffold size substantially enhances their MLGI affinity and antiviral potency.

## 1. Introduction

Viral infections still pose a significant global health concern. Particularly, the recent COVID-19 pandemic, which has caused the death of tens of millions worldwide [1,2] has once again highlighted the critical importance of gaining a comprehensive understanding of virus–host interactions. The process of viruses attaching, entering, and initiating the infection of host cells relies on complex multivalent interactions between viral glycoproteins and host cell surface receptor proteins, or vice versa. Glycans, complex carbohydrate molecules, are essential biological components found on the cell surface that enable pathogen–host cell contacts [3,4]. Most pathogens typically engage with carbohydrate-binding proteins, known as lectins, on target cells via their surface glycans, allowing them to penetrate through the plasma membrane of the host cells to initiate infection [5,6,7,8]. Multivalent lectin–glycan interactions (MLGIs) enable the binding of multiple lectin sites to a dense array of target glycans on the cell surface, significantly enhancing the binding affinity and specificity [4,9,10]. MLGIs also allows viruses to overcome the typically low-affinity nature of individual monovalent lectin–glycan interactions (with binding dissociation constant (*K*_d_) typically in the range of µM-mM) [3,11], thereby ensuring a stable and robust attachment to host cells. However, some MLGIs remain poorly understood, particularly for host cell lectins with multiple, flexibly presented carbohydrate recognition domains (CRDs) that are widely involved in viral glycan interactions. Therefore, elucidating the structural and binding mechanisms of such MLGIs not only can provide valuable insights into how viruses utilise MLGIs to establish infection and evade immune responses but also allows us to design spatially matched multivalent glycan-based inhibitors to potentially target specific MLGIs for therapeutic purposes, which can also offer a promising approach for new antiviral therapies.

The design and development of potent glycoconjugates to block viral binding to host cells can offer a distinct advantage over conventional antiviral strategies, especially for controlling infections as they can inhibit viral activity without the risk of inducing resistance development [12,13,14]. Free monovalent glycans can be used for this purpose, but their efficiency is rather low, due to their weak monovalent interaction with target lectins. Glycoconjugates can address this limitation by establishing multivalent interactions with target lectins’ multiple CRDs. This often results in substantially increased binding strength—by 5 to 7 orders of magnitude compared to their corresponding monovalent binding [11,15]—greatly enhancing their potency in inhibiting viral adhesion and infection of host cells. In this context, nanomaterials can serve as effective scaffolds for displaying multivalent sugar ligands, thereby enabling robust and specific targeting of lectins. Gold nanoparticles (GNPs) are excellent scaffolds for creating an effective, polyvalent glycan nanoplatform owing to their distinctive properties. Their high surface area-to-volume ratio allows for dense and stable polyvalent display of glycans in three dimensions. Their great colloidal stability ensures that GNP-glycans remain intact as individual particles within biological environments without aggregation [16,17,18]. Moreover, a change in GNP-glycan assembly or clustering following lectin binding can be conveniently monitored by simple methods such as UV-visible spectroscopy or by measuring their hydrodynamic sizes [14,19,20,21]. This provides a reliable and straightforward method for evaluating binding interactions, thereby enhancing our understanding of glycan–lectin dynamics in viral inhibition. The development of GNP-based glycan-nanoplatforms signifies a novel paradigm in the prevention and treatment of viral infections. This approach may contribute to the prevention of infection processes by potent and specific targeting of lectin-based virus receptors. A deeper understanding of MLGI dynamics can facilitate the integration of such innovative strategies into new antiviral therapies, potentially leading to an effective solution to address viral resistance development.

Recently, we have pioneered the use of glycan-nanoparticles as novel biophysical probes for investigating MLGI mechanisms by fully exploiting nanoparticles’ unique size-dependent chemical and physical properties (e.g., nanoscale size, high TEM contrast, strong fluorescence for quantum dot or strong fluorescence quenching for GNPs) [14,19,20,21,22,23]. We functionalised the GNPs with lipoic acid-oligo(ethylene glycol) based multifunctional ligand terminating with α-mannose (Man) and α-manno-α-1,2-biose (DiMan) to make GNP-Man and GNP-DiMan conjugates, subsequently used them to study their interactions with a pair of model tetrameric lectins, a dendritic cell surface lectin (DC-SIGN) and an endothelial cell lectin (DC-SIGNR), through the fluorescence quenching technique [14,21]. These two lectins (collectively referred to DC-SIGN/R, hereafter) are crucial for mediating and/or facilitating the infection of many viral pathogens, including HIV, HCV, Zika, SARS-CoV-2, and Ebola virus [13,24,25,26,27,28]. Despite almost identical tetrameric architecture and monovalent Man-CRD-binding motifs, they show different virus-binding and transmitting properties. For instance, DC-SIGN is more efficient than DC-SIGNR in transmitting the HIV infection, while only DC-SIGNR, but not DC-SIGN, can transmit the West Nile Virus infection [19,28,29]. The biophysical mechanisms underlying their differences in viral transmissions remain not fully understood. Using GNPs of varying sizes polyvalently coated with Man- and DiMan-based natural glycan ligands onto their surfaces, we have developed a new GNP-fluorescence quenching affinity assay for MLGIs. We showed that the MLGI affinity between GNP-glycan and DC-SIGN/R increased significantly with the increasing GNP size, suggesting that a larger GNP scaffold provides the glycan ligands with more favourable positional and spatial conformations for forming strong MLGIs with DC-SIGN/R. Moreover, substituting the GNP surface terminal glycan from Man to DiMan resulted in a marked increase in their MLGI affinity with DC-SIGN/R [14,21], attributed to DiMan’s ability to engage with both the primary and secondary binding sites of DC-SIGN/R’s CRDs [7]. Further, the binding modes between DC-SIGN/R and GNP-glycans were revealed via a combined analysis of the hydrodynamic diameter (*D*_h_) and TEM images of the resulting GNP-glycan-DC-SIGN/R assemblies captured in their native dispersion states. We found that all four CRDs in each DC-SIGN bind simultaneously to only one GNP-glycan, whereas each DC-SIGNR molecule cross-links with different GNP-glycans [14]. The different binding modes stem from the distinct orientations of their four CRDs in DC-SIGN/R. These GNP-glycans also inhibited entry of particles pseudotyped with the Ebola virus glycoprotein (EBOVpp) into DC-SIGN- or DC-SIGNR expressing host cells, with impressively low *EC*_50_ values down to ~23 pM and ~49 pM, respectively, making them the most potent glycoconjugate inhibitors against EBOVpp entry in this cellular model [21]. Despite such significant results, how different glycan ligand types on the GNP surface affect their MLGI affinity and antiviral properties remains to be explored.

Herein, we extend the glycan ligand on our GNP-glycan probes to a synthetic mannuronic acid (ManA)-based ligand (Figure 1). The only structural difference between Man, a natural glycan ligand of DC-SIGN/R, and ManA is that the -CH_2_OH group at position 6 in Man is now replaced by a -CO_2_H group, while the two -OH groups at positions 3 and 4 (mainly responsible for binding to DC-SIGN/R via coordination to the Ca^2+^ ion in their CRD glycan binding site [7]) are the same. This modification might enhance its binding with DC-SIGN/R CRD by forming a new Ca^2+^ coordination, hydrogen bonding, and/or electrostatic interactions. We polyvalently coated two different sized GNPs (e.g., ∼5 and ∼13 nm in diameter, denoted as G5 and G13, respectively) with the ManA ligand to investigate how this controls their MLGI parameters with DC-SIGN/R. GNP’s strong inner filter effect (arising from their strong plasmonic absorption) could still cause significant interference in fluorescence quenching measurements, despite the selection of Atto-643, a far-red emitting fluorophore with a red excitation wavelength, λ_EX_ = 630 nm. To minimise this effect, we performed the GNP-fluorescence quenching assay under a fixed GNP-glycan concentration but with varying protein concentrations. In this way, the MLGI affinities of both small and large GNP-glycan conjugates with DC-SIGN/R could be accurately measured by our previously reported GNP based fluorescence quenching method [21]. Our results indicate that both G5 and G13 capped with the LA-EG_2_-ManA ligand (see Figure 1 for chemical structure, denoted as Gx-ManA, x = 5 and 13) exhibit strong MLGI affinities with the DC-SIGN/R which are enhanced with the increasing GNP scaffold size. Furthermore, we also report that DC-SIGN/R-dependent EBOVpp (vesicular stomatitis virus (VSV) particles bearing the Ebola virus glycoprotein) entry into host cells is potently blocked by Gx-ManA conjugates, with potency being enhanced with increasing GNP scaffold size. Notably, G13-ManA blocks both DC-SIGN/R-mediated virus infections with sub-nM *EC*_50_ values. This work suggests that Gx-ManA conjugates can be used as multifunctional probes for MLGI affinity and potential antiviral agent by blocking viral infections.

## 2. Experimental Section

### 2.1. Materials

All chemicals were purchased from Acros Organics (Geel Belgium), Alfa Aesar (Ward Hill, MA, USA), Biosynth (Staad, Switzerland), Fisher Scientific (Waltham, MA, USA), Fluorochem (Derbyshire, UK), Sigma Aldrich (St. Louis, MO, USA), TCI Chemical (Tokyo, Japan), and Thermo Scientific (Waltham, MA, USA) with >99% impurity and used as received without further purification unless specified elsewhere. NH_2_-EG_2_-C≡CH was purchased from PurePEG LLC (San Diego, CA, USA). Ultrapure water (resistance > 18.2 MΩ·cm) purified by an ELGA Purelab classic UVF system was used for all experiments and making all buffers.

### 2.2. Synthesis of [2-(2-Azidoethoxy)ethoxy]-α-d-Mannopyranosiduronic Acid (N_3_-EG_2_-ManA] [30,31,32]

#### 2.2.1. Phenyl 2,3,4,6-Tetra-O-Acetyl-1-Thio-α-d-Mannopyranoside (**1**)

To a stirred solution of α-d-mannose pentaacetate (22.2 g, 56.9 mmol, 1.0 equiv.) in DCM (55 mL), thiophenol (8.7 mL, d = 1.078, 85.3 mmol, 1.5 equiv.) and BF_3_·Et_2_O (33.7 mL, 273 mmol, 4.8 equiv.) were added successively. The reaction mixture was then stirred at RT for 48 h, then washed successively with sat. aq. NaHCO_3_ (4 × 80 mL), 5% aq. NaOH solution (5 × 70 mL) and brine (3 × 50 mL). The organic phase was dried over MgSO_4_, filtered and the solvent was removed in vacuo. The residue, a pale-yellow oil, was crystallised from EtOH (~80 mL) to give Compound **1** as a white powder (18.9 g, 43.0 mmol, 75%). R_f_ 0.73 (EtOAc/hexane, 1/2). m.p. 86–88 °C. ^1^H NMR (300 MHz; CDCl_3_) δ 7.50–7.30 (m, 5H, Ar-H), 5.49–5.51 (m, 2H, H1, H3), 5.35–5.32 (m, 2H, H2, H4), 4.57–4.53 (m, 1H, H5), 4.31 (dd, *J* = 12.2, 5.7 Hz, 1H, H6a), 4.10 (dd, *J* = 12.2, 1.5 Hz, 1H, H6b), 2.16 (s, 3H, C(O)CH_3_), 2.08 (s, 3H, C(O)CH_3_), 2.06 (s, 3H, C(O)CH_3_), 2.03 (s, 3H, C(O)CH_3_). ^13^C NMR (101 MHz; CDCl_3_) δ 170.5 (C=O), 169.9 (C=O), 169.8 (C=O), 169.7 (C=O), 132.7 (Ar-C), 132.1 (Ar-CH), 129.2 (Ar-CH), 128.1 (Ar-CH), 85.7 (C1), 70.9 (C3), 69.6 (C5), 69.4 (C2), 66.4 (C4), 62.5 (C6), 20.9 (C(O)*C*H_3_), 20.7 (C(O)*C*H_3_), 20.7 (C(O)*C*H_3_), 20.6 (C(O)*C*H_3_). LRMS (ES^+^) *m/z* 458 [(M+NH_4_)^+^, 100%]; HR-MS *m/z* (ESI^+^) Found (M+NH_4_)^+^ 458.1475, C_20_H_24_O_9_SNH_4_ requires 458.1479.

#### 2.2.2. 2,3,4,5-Tetra-O-Acetyl-α-d-[2-(2-Azidoethoxy)ethoxy]mannopyranoside (**2**)

A solution of the thioglycoside (**1**) (1.30 g, 2.95 mmol, 1.0 equiv.) and 2-[2-(2-chloroethoxy)ethoxy]ethanol (1.5 mL, 10.33 mmol, 3.5 equiv.) in DCM (24 mL) was stirred over 4 Å M.S. for 30 min before NIS (863 mg, 3.83 mmol, 1.3 equiv.) was added. The mixture was cooled to −50 °C before TMSOTf (0.16 mL, 0.88 mmol, 0.3 equiv.) was added. The solution was gradually warmed up to 0 °C over 6 h, and upon completion, Et_3_N (5.0 equiv.) was added until pH = 7. The reaction mixture was washed with 10% aq. Na_2_S_2_O_3_ solution, brine and concentrated in vacuo. The crude product was purified via manual flash column chromatography (1.5/1, 1/1, 1/2 EtOAc/Petroleum ether) to furnish 2,3,4,5-tetra-*O*-acetyl-α-d-[2-(2-chloroethoxy)ethoxy] mannopyranoside as a colourless oil (920 mg, 1.86 mmol, 63%). R_f_ 0.27 (EtOAc/petroleum ether, 1/1). ^1^H NMR (400 MHz, CDCl_3_) δ 5.36 (dd, *J* = 10.0, 3.4 Hz, 1H, H3), 5.32–5.26 (m, 2H, H2, H4), 4.88 (d, *J* = 1.7 Hz, 1H, H1), 4.32–4.26 (m, 1H, H6a), 4.13–4.05 (m, 2H, H5, H6b), 3.79–3.61 (m, 12H, 6 × CH_2_), 2.16 (s, 3H, C(O)CH_3_), 2.11 (s, 3H, C(O)CH_3_), 2.05 (s, 3H, C(O)CH_3_), 1.99 (s, 3H, C(O)CH_3_). ^13^C NMR (101 MHz, CDCl_3_) δ 170.7 (C=O), 170.1 (C=O), 169.9 (C=O), 169.8 (C=O), 97.7 (C1), 71.4 (CH_2_), 70.7 (CH_2_), 70.3 (CH_2_), 70.0 (CH_2_), 69.6 (C2), 69.1 (C3), 68.4 (C5), 67.4 (CH_2_), 66.2 (C4), 61.7 (C6), 42.7 (CH_2_Cl), 20.9 (C(O)*C*H_3_), 20.8 (C(O)*C*H_3_), 20.7 (C(O)*C*H_3_), 20.7 (C(O)*C*H_3_).

Then, 2,3,4,5-Tetra-*O*-acetyl-α-d-[2-(2-chloroethoxy)ethoxy]manno-pyranoside (920 mg, 1.84 mmol, 1.0 equiv.), NaN_3_ (600 mg, 9.22 mmol, 5.0 equiv.) and TBAI (2.72 g, 7.37 mmol, 4.0 equiv.) were dissolved in DMF (18 mL) then stirred for 24 h at 80 °C. Upon completion, the reaction mixture was cooled to RT and EtOAc (25 mL) was added. The organic layer was washed with H_2_O, brine, dried over MgSO_4_, filtered and concentrated in vacuo to afford the crude product. Purification using silica gel flash column chromatography (1/2, 1/1 EtOAc/petroleum ether) afforded the title Compound **2** as a colourless oil (680 mg, 1.34 mmol, 73%). R_f_ 0.26 (EtOAc/hexane, 1/2). ^1^H NMR (400 MHz, CDCl_3_) δ 5.36 (dd, *J* = 10.1, 3.4 Hz, 1H, H3), 5.32–5.26 (m, 2H, H2, H4), 4.88 (d, *J* = 1.7 Hz, 1H, H1), 4.29 (dd, *J* = 12.1, 4.9 Hz, 1H, H6a), 4.13–4.04 (m, 2H, H5, H6b), 3.76–3.72 (m, 4H, 2 × CH_2_), 3.64–3.61 (m, 6H, 3 × CH_2_), 3.40 (s, 2H, CH_2_N_3_), 2.16 (s, 3H, C(O)CH_3_), 2.11 (s, 3H, C(O)CH_3_), 2.04 (s, 3H, C(O)CH_3_), 1.99 (s, 3H, C(O)CH_3_). ^13^C NMR (101 MHz, CDCl_3_) δ 170.7 (C=O), 170.1 (C=O), 169.9 (C=O), 169.8 (C=O), 97.7 (C1), 70.8 (CH_2_), 70.7 (CH_2_), 70.4 (CH_2_), 70.1 (CH_2_), 69.6 (C2), 69.1 (C3), 68.4 (C5), 67.4 (CH_2_), 66.2 (C4), 62.4 (C6), 50.7 (CH_2_N_3_), 20.9 (C(O)*C*H_3_), 20.8 (C(O)*C*H_3_), 20.7 (C(O)*C*H_3_), 20.7 (C(O)*C*H_3_).

#### 2.2.3. 2,3,4,5-Tetra-O-Hydroxy-α-d-[2-(2-Azidoethoxy)ethoxy]mannopyranoside (**3**)

To a stirred solution of the azide (**2**) (450 mg, 0.89 mmol, 1.0 equiv.) in MeOH (8 mL) was added NH_3_ (0.25 mL, 7 M in MeOH, 1.78 mmol, 2.0 equiv.) and Et_3_N (0.12 mL, 0.89 mmol, 1.0 equiv.) then the reaction mixture was stirred for 48 h at RT. Upon completion, the reaction was neutralised with ion exchange Amberlite 120 (H^+^) resin (approximately 0.2 g, 5 min), filtered, and concentrated under reduced pressure. Flash column chromatography (5% → 15% MeOH in DCM) afforded the Compound **3** as a colourless oil (270 mg, 0.80 mmol, 89%). R_f_ 0.90 (15% MeOH in DCM). ^1^H NMR (400 MHz, CDCl_3_) δ 4.87 (d, *J* = 1.5 Hz, 1H, H1), 3.98–3.90 (m, 3H, H2, H4, H6a), 3.85 (dd, *J* = 9.4, 3.3 Hz, 1H, H3), 3.78 (dtd, *J* = 8.6, 5.5, 2.1 Hz, 2H, H6b, CH_2_), 3.71–3.61 (m, 9H, CH_2_), 3.57 (dd, *J* = 9.6, 2.2 Hz, 1H, H5), 3.40 (dd, *J* = 5.6, 4.5 Hz, 2H, CH_2_N_3_). ^13^C NMR (101 MHz, CDCl_3_) δ 100.2 (C1), 72.3 (C5), 71.6 (C3), 70.9 (C2), 70.7 (CH_2_), 70.7 (CH_2_), 70.2 (CH_2_), 70.1 (CH_2_), 66.8 (CH_2_), 66.3 (C4), 61.1 (C6), 50.7 (CH_2_N_3_).

#### 2.2.4. [2-(2-Azidoethoxy)ethoxy] α-d-mannopyranosiduronic Acid (**4**)

To a vigorously stirred solution of (**3**) (120 mg, 0.41 mmol, 1.0 equiv.) in DCM/H_2_O (1/1, 4 mL) was added TEMPO (60 mg, 0.41 mmol, 1.0 equiv.) and BAIB (1.33 g, 4.15 mmol, 10.0 equiv.). Stirring was continued for 16 h, and the reaction mixture was diluted with H_2_O (10 mL) and washed with DCM (10 mL). The aqueous phase was concentrated under reduced pressure then the crude product was purified by C-18 column chromatography (10% MeCN in H_2_O) and lyophilised to afford Compound **4** as a white foam (108 mg, 0.368 mmol, 90%). R_f_ 0.85 (H_2_O/MeCN, 9/1). ^1^H NMR (400 MHz, D_2_O) δ 4.81 (d, *J* = 1.8 Hz, 1H, H1), 3.89 (dd, *J* = 3.5, 1.8 Hz, 1H, H2), 3.83–3.78 (m, 2H, H6a, CH_2_), 3.77–3.73 (m, 1H, H3), 3.66 (d, *J* = 5.7 Hz, 10H, H6b, CH_2_), 3.59–3.56 (m, 2H, H4, H5), 3.43 (dd, *J* = 5.6, 4.2 Hz, 2H, CH_2_N_3_). ^13^C NMR (101 MHz, D_2_O) δ 99.9 (C1), 72.7 (C5), 70.5 (C3), 69.9 (C2), 69.6 (CH_2_), 69.6 (CH_2_), 69.5 (CH_2_), 69.3 (CH_2_), 66.7 (CH_2_), 66.4 (C4), 60.9 (C6), 50.2 (CH_2_N_3_). HR-MS *m/z* (NSI^−^) found (M-H)^−^ 350.1196, C_12_H_20_O_9_N_3_ requires 350.1200.

### 2.3. Synthesis of LA-EG_2_-ManA [14,19,20,21]

LA-EG_2_-C≡CH was synthesised amide coupling between lipoic acid (LA) and commercial NH_2_-EG_2_-C≡CH as described previously [14,19,20,21]. LA-EG_2_-C≡CH (15 mg, 0.045 mmol), N_3_-EG_2_-ManA (17 mg, 0.048 mmol), CuSO_4_·5H_2_O (0.4 mg, 0.0016 mmol), TBTA (1.5 mg, 0.0028 mmol), and sodium ascorbate (1.2 mg, 0.006 mmol) were dissolved in 2 mL of THF/H_2_O (1:1, vol/vol) to allow for efficient click reaction between LA-EG_2_-C≡CH and N_3_-EG_2_-ManA. The resulting solution was stirred overnight at RT in darkness. The next day, the consumption of all starting compounds was confirmed by TLC. The solvent was then evaporated, and the desired ligand was purified by size exclusion chromatography using a Biogel P2 column using ammonium formate as an eluent to obtain the pure product, LA-EG_2_-ManA, in 72% yield.

TLC: (CHCl_3_/MeOH 1:1) R_f_ 0.62; ^1^H-NMR (500 MHz, D_2_O) δ (ppm): 8.44 (s, 1H), 8.10 (s, 1H), 4.90 (d, 1H), 4.70 (s, 2H), 4.65 (t, 2H), 3.99 (t, 2H), 3.85 (m, 5H), 3.69 (m, 12H), 3.61 (t, 2H), 3.37 (t, 2H), 3.20 (m, 2H), 2.47 (m, 1H), 2.24 (t, 2H), 1.96 (m, 1H), 1.72 (m, 1H), 1.60 (m, 3H), 1.38 (m, 2H); ^13^C-NMR (125 MHz, D_2_O) δ (ppm): 176.9 (C=O), 170.9, 143.8 (C=CH), 125.4 (C=CH), 99.9, 72.8, 70.3, 69.8, 69.6, 69.4, 69.4, 69.4, 68.9, 68.8, 68.7, 66.6, 63.0, 56.4, 50.1, 50.0, 40.2, 38.8, 38.0, 35.4, 33.6, 27.7, 24.9; LC-MS: calculated *m/z* for C_27_H_46_N_4_O_12_S_2_ (M+H)^+^ 683.26; found 683.22.

### 2.4. Preparation of Gx-ManA Conjugates [21]

G5-ManA was prepared by mixing G5 and LA-EG_2_-ManA ligand via self-assembly in an aqueous solution. G5s (6 mL, 91 nM) suspended in citrate solution were concentrated to 250 μL using a 30 K MWCO spin column and washed with H_2_O (3 × 250 μL) to remove any unbound impurities. The resulting concentrated G5 solution was then directly mixed with LA-EG_2_-ManA ligand at a G5: ligand molar ratio of 1:1000. The resulting mixture was left stirring at RT in darkness overnight to form G5-ManA conjugates. After that, the mixture was transferred to a 30 K MWCO centrifugal filter and centrifuged at 4000 rpm for 20 min, and the G5-ManA residues were washed with H_2_O (3 × 250 μL) to remove any unbound free ligands and then dispersed in pure water to make the G5-ManA stock.

For the preparation of G13-ManA, 20 mL each of the citrate stabilised G13 stock solution was directly added to the required amount of LA-EG_2_-ManA ligand stock solution in water at a GNP: ligand molar ratio of 1:3000. The resulting solution was stirred at RT overnight in the dark to make the G13-ManA conjugates via gold-thiol self-assembly. After that, the resulting mixtures were divided into 1.5 mL portions in Eppendorf tubes and centrifuged at 17,000× *g* for 30 min to remove any unbound free ligands. After the careful withdrawal of the clear supernatant, the G13-ManA residues were washed with pure water (3 × 500 μL), followed by centrifugation and washing with water three times to remove any unbound free ligands. The *D*_h_ histograms for G5- and G13-ManA were given in Figure 2. Their concentrations were determined by the Beer-Lambert law using their molar extinction coefficients of 1.10 × 10^7^ M^−1^cm^−1^ for G5- and 2.32 × 10^8^ M^−1^cm^−1^ for G13-ManA [21].

All the filtrate and washing-through liquids were collected, combined, freeze-dried, and then re-dissolved in 1.40 mL pure water to determine the amount of unbound LA-EG_2_-ManA ligand using the phenol-sulphuric acid method described previously [14,19,21]. A total of 25 μL of each solution was diluted with water to a final volume of 125 μL. This solution was then mixed with 125 μL of 5% phenol and 625 μL of H_2_SO_4_ (98%) and then allowed to incubate at RT for 30 min. The absorption of the solution at 490 nm was recorded, and the dilution factors were then corrected to calculate the total amount of unconjugated glycan ligand against a standard calibration curve obtained with pure LA-EG_2_-ManA ligand. The difference in LA-EG_2_-ManA ligand amount between that added and that remained in the supernatant was considered to have conjugated onto the GNP surface [14,19,21].

### 2.5. Fluorescence Spectra [21]

All fluorescence spectra were recorded on a Horiba FluoroMax-4 Spectrofluorometer using a 0.70 mL quartz cuvette under a fixed excitation wavelength (λ_ex_) of 630 nm. Emission spectra over 650–800 nm were collected with excitation and emission slit widths of 5 nm under a slow scan speed. All measurements were carried out in a binding buffer (20 mM HEPES, pH 7.8, 100 mM NaCl, 10 mM CaCl_2_) containing 1 mg/mL BSA to reduce any non-specific interactions and absorption to cuvette walls. The required amounts of Gx-DiMan and DC-SIGN/R were mixed and then incubated at room temperature (RT) for 20 min before recording their fluorescence spectra. The fluorescence spectra from 643 to 800 nm were integrated and used to calculate the quenching efficiency (QE).

### 2.6. Virus Inhibition Studies

The inhibition effects of Gx-ManA conjugates were evaluated using our established procedures [14,19,21]. Briefly, 293T cells seeded in 96-well plates were transfected with plasmids encoding DC-SIGN/R or control transfected with empty plasmid (pcDNA). Culture medium, Dulbecco’s Modified Eagle Medium (DMEM) containing 10% fetal bovine serum (FBS), was replaced by fresh medium at 16-h post transfection and the cells were further cultivated at 37 °C and 5% CO_2_. At 48 h post-transfection, the cells were exposed to twice the final concentration of Gx-DiMan inhibitor in the OptiMEM-medium for 30 min in a total volume of 50 μL. After that, the resulting cells were inoculated with 50 μL of preparations of the luciferase gene encoding vesicular stomatitis virus (VSV) vector particles bearing either EBOV-GP which can use DC-SIGN/R for the augmentation of host cell entry or the vesicular stomatitis virus glycoprotein (VSV-G). The latter cannot use DC-SIGN/R to increase host cell entry and is thus employed as a specificity control. Under these conditions, the binding of Gx-ManA particles to 293T cell surface DC-SIGN/R receptors can block EBOV-GP interactions with such lectin receptors, reducing the virus particle transduction efficiency and thus producing reduced cellular luciferase activity. At 16–20 h post-infection luciferase activities in cell lysates were determined using a commercially available kit (PJK), following the manufacturer’s instructions, as described in our previous publications [14,19,21].

### 2.7. Cytotoxicity Studies (Cell Titer-Glo Assay)

For analysis of the potential cellular toxicity of Gx-ManA, the Cell Titer-Glo Luminescent Cell Viability Assay (Promega, Singapore) was used following the manufacturer’s instructions. This assay measures ATP levels in cells as a marker for cell viability with light emission as readout. In brief, 293T cells seeded in 96-well plates were incubated at 37 °C, 5% CO_2_, for 24 h in the presence or absence of Gx-ManA. Thereafter, the cells were washed with PBS, exposed to assay substrate and incubated for 2 min on an orbital shaker, followed by quantification of light emission using a Hidex Sense plate luminometer. Cells exposed to 10% Triton X served as positive control for cytotoxicity. No apparent cytotoxicity was detected even at the highest Gx-ManA doses used in antiviral studies (see Appendix A).

### 2.8. Data Analysis and Fitting [14,21]

All fluorescence and DLS data were analysed using the Origin software (version 2025). The fluorescence spectra of lectins alone and lectin + Gx-ManA samples were integrated and used to calculate the QEs and presented as mean ± standard errors (SEs). The (QE × *C*) vs. *C* plots were fitted by the linear function, accounting for the SEs of each data point, to give the best fits (highest *R*^2^ values). The DLS histograms were fitted by the standard single Gaussian function to obtain the *D*_h_ and full-width at half-maximum (FWHM, shown as W) and depicted in each DLS graph. The results obtained from the best fits were listed in the relevant tables with the standard fitting errors.

## 3. Results and Discussion

### 3.1. Ligand Design and Synthesis

To probe how mannuronic acid (ManA)-functionalised GNPs control their MLGI properties, a lipoic acid-di (ethylene glycol)-ManA (LA-EG_2_-ManA) ligand was synthesised via our established protocols as shown schematically in Figure 1 [14,19,20,21,22,23,33]. This ligand was designed to have three distinct functional domains [19,34]: a lipoic acid (LA) anchoring domain for strong chelating onto the GNP surface through the formation of two stable Au-S bonds [14,21], a hydrophilic and flexible EG_2_ linker domain for enhancing the water solubility and dispersibility, and providing resistance to non-specific interactions [35,36], and a terminal ManA group for lectin binding.

The synthesis route for the LA-EG_2_-ManA ligands is outlined in Figure 1. First, an LA-EG_2_-C≡CH linker was synthesised using the standard dicyclohexylcarbodiimide/N,N-dimethyl-aminopyridine (DCC/DMAP)-mediated amide coupling between lipoic acid and H_2_N-EG_2_-acetylene [14]. Then, [2-(2-azidoethoxy)ethoxy]-α-d-mannopyranosiduronic acid, N_3_-EG_2_-ManA, was synthesised using the procedure described in the experimental section (see Appendix A for spectral characterisation). Finally, the LA-EG_2_-C≡CH linker was covalently linked to N_3_-EG_2_-ManA through Cu-catalysed click chemistry, yielding the desired LA-EG_2_-ManA ligand in good yields. Its chemical structure and purity were verified through ^1^H/^13^C-NMR and LC-MS spectra (see Appendix A for ^1^H/^13^C-NMR and LC-MS spectra).

### 3.2. Preparation of GNP–ManA Conjugates

In this study, we used two previously synthesised GNPs with average core diameters of ∼5 and ∼13 nm, hereafter referred as G5 and G13, respectively, as scaffolds to make the Gx-ManA conjugates [21]. This was achieved by incubating citrate-stabilised GNPs overnight with the LA-EG_2_-ManA ligands in water at a fixed ligand-to-GNP molar ratio of 1000:1 for G5 and 3000:1 for G13. We previously found that GNP-glycans produced using LA-glycan ligands and their reduced dihydrolipoic acid (DHLA)-forms have the same *D*_h_ value and stability. This indicated that the LA-based ligands could directly self-assemble onto the GNP surface efficiently without reduction, allowing us to directly use such air-stable LA-glycan ligands to prepare GNP-glycan conjugates [14,21]. The resulting Gx-ManA conjugates were purified by ultrafiltration using 30 K MWCO filter tubes for G5-ManA or centrifugation for G13-ManA, followed by washing with pure water as described previously. The unbound free ligands in the supernatant and washings were collected to determine Gx surface glycan valency. The Gx-ManA conjugates were found to completely resist NaCl (250 mM) induced aggregation, indicating that their surfaces were successfully functionalised with the desired hydrophilic glycan ligands (citrate-stabilised GNPs readily aggregate in a moderate NaCl content, due to NaCl screening of the electrostatic repulsions among the negatively charged GNPs) [14,21]. The citrate-stabilised G5 and G13 both exhibited single Gaussian distribution species with a hydrodynamic diameter (*D*_h_, based on volume population) of ~7 and ~16 nm, respectively [21]. After coating with the LA-EG_2_-ManA ligands, the resulting Gx-ManA conjugates were found to be uniform and monodisperse, both in pure water and in a binding buffer (20 mM HEPES, 100 mM NaCl, 10 mM CaCl_2_, pH 7.8), with *D*_h_s of ~12 and ~22 nm for G5- and G13-ManA, respectively (Figure 2). These *D*_h_s were a few nm larger than those of their corresponding citrate-stabilised Gxs, consistent with those expected for single Gx particles coated with a monolayer of the LA-EG_2_-ManA ligand, which is longer than the native citrate ligands.

Interestingly, G5-ManA has a similar *D*_h_ size to a gp160 trimer, the HIV surface densely glycosylated spike protein responsible for mediating HIV-DC-SIGN interactions and viral infection [37,38]. Thus, it was selected to mimic gp160-DC-SIGN interactions. The larger G13-ManA was employed to investigate the impact of GNP scaffold size on their MLGI properties with DC-SIGN/R. The Gx-ManA solutions were highly stable, with no changes in physical appearance or precipitation after extended storage at 4 °C for >12 months. The glycan valency (number of LA-EG_2_-ManA ligands) per Gx particle was estimated from the ligand amount difference between the added and the remaining unbound in the supernatant after GNP conjugation using a phenol-sulfuric acid carbohydrate quantification method [19,21,39], yielding a value of ∼530 for G5-ManA and ~1800 for G13-ManA (see Appendix A). This gave an LA-EG_2_-ManA ligand conjugation efficiency of ~53% and ~60% for G5 and G13, respectively, under our experimental conditions.

The average inter-glycan distance (*d*) was calculated based on their *D*_h_s value and glycan valency using the method reported previously [14,19,21,40,41], giving *d* values of ~1.01 nm for G5-ManA and ~1.08 nm for G13-ManA (see Appendix A). Interestingly, such *d* values align well with the main inter-glycan sequon spaces (e.g., 0.7–1.3 nm) found on the HIV surface glycoprotein gp160 trimer [38].

### 3.3. Quantifying GNP-Glycan-DC-SIGN/R Binding Affinity

GNPs have been shown to exhibit universal and strong fluorescence quenching properties against a range of different fluorophores [42]. In particular, its quenching efficiency (QE)-distance dependence was found to follow the nano-surface energy transfer (NSET) mechanism, unlike the Förster resonance energy transfer (FRET) mechanism exhibited by organic quenchers. In NSET [43,44], QE is inversely proportional to the 4th power of the distance, rather than the 6th power of conventional FRET, and thus, GNPs can offer considerably higher QE over a longer distance range than organic quenchers. This very useful property has made GNPs widely used in bioimaging, biosensing, and biodiagnostic research [45,46].

In our recent studies, we demonstrated that the superior fluorescence quenching features of GNPs can be employed as a new readout method to quantify MLGI affinities [14,19,21]. In this context, we introduced a site-specific cysteine mutation on the CRD of DC-SIGN/R molecules. The mutation site was selected at a position close to, but not in, their glycan binding pocket, ensuring that mutation does not adversely affect the CRD’s glycan binding properties. This was experimentally confirmed that the resulting mutant proteins retained the tetrameric structure and glycan binding properties of the native tetrameric DC-SIGN/R [14,19,21,33]. In this study, the mutant DC-SIGN/R proteins were labeled with maleimide-modified Atto-643 fluorophore, with >80% labeling efficiency per monomer, as reported previously [21]. Since the absorption extinction coefficient of GNPs increases roughly linearly with its volume, the strong absorption of large GNPs (e.g., ε = 2.32 × 10^8^ M^−1^cm^−1^ for G13) may interfere with fluorescence quenching measurements. Here, a fixed concentration (*C*) of 10 and 4 nM for G5 and G13, respectively, while varying DC-SIGN/R concentrations were employed. To further minimise any possible interference from GNP’s inner filter effect [21,47], the fluorescence spectra of the lectins (varying *C*s) without and with a fixed concentration of Gx-ManA were recorded at λ_EX_ = 630 nm, where Gxs have minimum absorption. All binding experiments were performed in a binding buffer containing 1 mg/mL bovine serum albumin (BSA) to provide an environment similar to biological conditions. This approach greatly reduced the adsorption of proteins and GNPs onto surfaces and nonspecific interactions by preventing non-specific interactions, which can be a major source of experimental errors at low *C*s, e.g., <10 nM [14,21]. Fluorescence spectra showing the binding of a fixed concentration of G5-ManA (10 nM) or G13-ManA (4 nM) with varying concentrations of DC-SIGN/R (from 0–80 nM) are given in Appendix A and their corresponding (QE × *C*) vs. *C* plots are summarised in Figure 3. The surface area of each G5-ManA and G13ManA particle is calculated to be ~430 and ~1490 nm^2^, respectively, from their *D*_h_s. Using a binding footprint area of ~35 nm^2^ per DC-SIGN molecule [19], we estimate that ~12 or ~43 copies of DC-SIGN molecule will be able to achieve saturate binding onto each G5-ManA or G13-ManA surface, respectively. The maximal lectin concentration here corresponds to a lectin: Gx-ManA molar ratio of 8:1 for G5ManA and 20:1 for G13ManA, which are both below those required to achieve saturate binding. We have found previously that below surface saturation, the QE of DC-SIGN/R binding to Gx-DiMan remained constant, allowing us to accurately derive QE from the slope of the corresponding (QE × *C*) vs. *C* plots [21]. In the absence of Gx-ManA, the fluorescence intensities of labeled DC-SIGN/R increased linearly with the increasing *C*, and their fluorescence was significantly reduced in the presence of Gx-ManA, consistent with that expected for binding-induced quenching of CRD labeled fluorophores by GNPs in proximity as shown schematically in Figure 1B. In this study, we used our previously reported method to quantitatively analyse the binding data. This is based on the calculation of the quenching efficiency (QE) of each (Gx-ManA + lectin) sample at certain protein concentrations using the following Equation (1) below:QE = (IF_0_ − IF)/IF_0_(1)
where IF_0_ and IF are the integrated protein fluorescence signals measured in the absence and presence of Gx-ManA, respectively. Since a GNP has been shown to quench fluorophores in close proximity by up to 99.97% [42], it is safe to assume that the observed QE reflects the proportion of added lectins that are bound to Gx-ManA [14,21].

As shown in our previous study, the QE of DC-SIGN, after mixing with a fixed amount of Gx-glycan, has remained almost constant over a certain range of protein: Gx molar ratio (PGR), but it decreased, as PGR was increased further, presumably due to surface saturation [21]. This result indicates that one DC-SIGN molecule bound to the Gx-glycan surface does not interfere with the binding of further DC-SIGN molecules on the same Gx-glycan, and a Gx-glycan bound to a few DC-SIGN molecules (before surface saturation) can produce the same level of QE as a free Gx-glycan. As a result, the binding equilibrium of multiple DC-SIGN molecules onto one Gx-ManA can be simplified to multiple 1:1 (i.e., one Gx-ManA to one DC-SIGN molecule) binding interactions by converting the binding into multiple 1:1 binding interactions, as previously described [21].DC-SIGN + Gx-glycan ↔ DC-SIGN-Gx-glycan(2)*K*_d_ = [DC-SIGN] [Gx-glycan]/[DC-SIGN-Gx-glycan](3)

Here [DC-SIGN], [Gx-glycan], and [DC-SIGN-Gx-glycan] are the equilibrium concentrations of free DC-SIGN, free Gx-glycan and bound DC-SIGN-Gx-glycan complex, respectively. For a reversible 1 to 1 interaction with equal starting concentration of *C*_0_ for both components, [DC-SIGN]_0_ = [Gx-glycan]_0_ = *C*_0_. As QE represents the portion of lectin bound to Gx-glycan, [DC-SIGN-Gx-glycan] = *C*_0_ × QE, and thus the equilibrium free [DC-SIGN] = [Gx-glycan] = *C*_0_ × (1 − QE). Taking these numbers into Equation (3) allows us to calculate the binding *K*_d_ via Equation (4):*K*_d_ = [*C*_0_ × (1 − QE)]^2^/(*C*_0_ × QE) = *C*_0_ × (1 − QE)^2^/QE(4)

To measure QE more accurately, a plot of (QE × *C*) vs. *C* (*C* = lectin concentration) over a lectin concentration range below that required to saturate Gx-ManA surface was employed to determine the average QE from linear fitting, where the slope obtained from the fit represents the average QE (Figure 3). For this method to work, the lectin concentration must be lower than that required to fully saturate the Gx-ManA surface. Otherwise, any excessively added lectin (not coming from the binding/dissociation equilibrium) will remain unbound and hence unquenched, leading to a reduced QE and the plot deviating significantly from linear function. The fitting parameters, QEs, and calculated *K*_d_s using Equation (4) for DC-SIGN/R binding with Gx-ManA were summarised in Table 1.

The *K*_d_s for Gx-ManA binding with DC-SIGN/R give two significant findings: (1) ManA-coated GNPs displayed relatively strong MLGI affinities for both DC-SIGN/R, with a slightly higher affinity toward DC-SIGNR. This result contrasts markedly with our previous findings [21], where GNPs coated with Man or DiMan, the natural mannose-based glycan ligands for DC-SIGN/R CRDs, showed markedly stronger binding to DC-SIGN over DC-SIGNR. Given that the CRDs in DC-SIGN/R possess the near-identical mannose-binding motifs (via coordination of the 3,4-hydroxyl groups to the Ca^2+^ ion in the binding sites), their affinity difference should originate from their differences in CRD orientation between DC-SIGN and DC-SIGNR. Indeed, our previous results indicated that all four CRDs in each DC-SIGN molecule are positioned uprightly, allowing it to bind tetravalently to one GNP-glycan, whereas those in each DC-SIGNR are split into two pairs and point sideways, allowing it to crosslink with multiple GNP-glycan particles [14,21]. Our result implies that replacing the -CH_2_OH (in Man) by a -CO_2_H group (in ManA) enhances its binding with DC-SIGNR. (2) The MLGI affinity between Gx-ManA and DC-SIGN/R was enhanced notably (5–6 folds) as the GNP scaffold size increased from 5 to 13 nm. For example, the *K*_d_s for Gx-ManA (x = 5 and 13) in binding to DC-SIGN were found to be ~61 and ~11 nM, while those for DC-SIGNR were ~48 and ~8 nM, respectively, indicating that a larger GNP scaffold provides a more favourable glycan display to enhance their MLGI affinity with DC-SIGN/R. This is likely due to the reduced surface curvature of the larger GNP over the smaller one creates a flatter display of surface glycans, which offers a better topological match for binding to all four CRDs in one DC-SIGN molecule to form simultaneous 1:1 (one Gx-ManA particle to one DC-SIGN molecule) tetravalent binding as shown schematically in Figure 1B. This result differs significantly from some other lectins reported in the literature, including ConA [48,49], most likely due to their very different binding modes, e.g., crosslinking (for ConA) vs. simultaneous 1:1 tetravalent binding (for DC-SIGN) [14,19]. Compared to crosslinking, the later binding mode is more thermodynamically favourable because not only can it maximise the MLGI favourable enthalpy changes (with full lectin binding site occupation) but also minimise the unfavourable entropic penalty (by forming small individual lectin-GNP-glycan complexes vs. largescale lectin-GNP-glycan assemblies). The dependence of DC-SIGN MLGI affinity on GNP scaffold size observed with Gx-ManA is fully consistent with our previous findings obtained with GNPs coated with both DC-SIGN’s natural glycan ligand, Gx-DiMan [21], and also a synthetic glycomimetic ligand, Gx-psDiMan [40]. Overall, our current findings suggest that increasing the size of the GNP scaffold is strongly beneficial for enhancing their MLGI affinity with both DC-SIGN and DC-SIGNR.

### 3.4. Inhibition of DC-SIGN/R-Promoted EBOVpp Entry into Cells

Considering their strong binding affinities with DC-SIGN/R (low- to mid- nM *K*_d_s), the Gx-ManA conjugates could act as potential inhibitors to block cell surface DC-SIGN/R-mediated viral infection. This ability was evaluated on our well-established cellular infection model [14,21]. The human embryonic kidney 293T cells transfected to express DC-SIGN or DC-SIGNR were used as model host cells and single-cycle vesicular stomatitis virus (VSV) particles pseudotyped with the Ebola virus glycoprotein (EBOV-GP, showing significant DC-SIGN/R augmented cell entry) and encoding the luciferase gene (abbreviated as EBOVpp) were used a model virus by following our previously established protocols [14,21]. The assay principle here is based on the competitive binding between GNP-glycans and viral particles to cell surface DC-SIGN/R receptors. Binding GNP-glycans to cell surface DC-SIGN/R should block these receptors from being able to further bind to viral surface EBOV-GP spikes, thereby inhibiting EBOVpp entry and reducing luciferase production in host cells. In addition, VSV particles carrying vesicular stomatitis virus glycoprotein (VSV-G), which do not rely on DC-SIGN/R for cell entry, were used as a control, to confirm whether the inhibition is specific to DC-SIGN/R-based MLGIs. Also, VSV particles encoding the luciferase gene, but lacking the viral glycoproteins, were employed as a negative control. All antiviral assays were conducted under standard DMEM cell culture medium supplemented with 10% fetal bovine serum (FBS) at 37 °C, as described previously [14,21]. The measured luciferase activities of 293T cell samples after incubation with various Gx-ManA conjugates are presented in Appendix A. Treatment with Gx-ManA led to a dose-dependent decrease in luciferase activity following EBOVpp infection. However, EBOVpp infection of control cells and infection driven by VSV-G was also affected (to a lesser degree), indicating that the antiviral action of Gx-ManA’s may be partially unspecific. The normalised viral inhibition data were fitted by a modified inhibitory model described in Equation (5).(5)$$NA=\frac{{{EC}_{50}}^{n}}{{{EC}_{50}}^{n}+C^{n}}$$
where *NA*, *EC*_50_, *C*, and *n* are the normalised luciferase activity, Gx-ManA concentration giving 50% apparent inhibition, Gx-ManA concentration, and inhibition coefficient, respectively [21]. The inhibition coefficient (*n*) indicates the cooperative nature of the inhibitory mechanism, where *n* < 1, = 1, or > 1 indicates a negative-, non-, or positive cooperativity. Here *n* describes the concentration-dependence of inhibitory behaviour, indicating how rapidly full inhibition can be achieved by increasing concentration. Where *n* ≥ 1 indicates a promising inhibitor, it can readily achieve complete inhibition by increasing concentration at reasonable levels, whereas *n* < 1 is less promising, as it is difficult to achieve full inhibition by simply increasing concentration [21]. The normalised luciferase activity, serving as an indicator of viral infection, was plotted against Gx-ManA concentration and fitted by Equation (5), and the results are depicted in Figure 4. Their corresponding fitting parameters are summarised in Table 2.

As shown in Table 2, both G5-ManA and G13-ManA displayed high potencies against DC-SIGN/R-promoted EBOVpp entry into 293T cells, as demonstrated by their sub- to low nM levels of *EC*_50_ values, although their antiviral specificity requires further investigation. Particularly, G13-ManA is significantly more promising, with *EC*_50_ of 0.53 ± 0.19 and 0.29 ± 0.12 nM against DC-SIGN- and DC-SIGNR- mediated EBOVpp entry, respectively. Such *EC*_50_ values are significantly lower than those of the G5-ManA counterpart. Therefore, the antiviral potency of Gx-ManA is significantly enhanced with the increasing GNP scaffold size. This result is not unexpected. In fact, it is fully consistent with their increased MLGI affinities for DC-SIGN/R measured by the GNP fluorescence quenching in solution described in the previous section (see Table 1). Interestingly, Gx-ManA exhibited comparable potencies (considering the errors in the *EC*_50_ values) against both DC-SIGN- and DC-SIGNR- promoted EBOVpp cellular entry. This result contrasts markedly from those of Gx-Man/DiMan, the same sized GNPs coated with DC-SIGN/R’s natural mannose-based glycan ligands, which were found to be more potent in inhibiting DC-SIGN- than DC-SIGNR- mediated EBOVpp infection [14,21]. Besides, Gx-ManA was found to show negative cooperativity (*n* < 1) in blocking both DC-SIGN- and DC-SIGNR- mediated viral infections. Again, this result differs significantly from those obtained with Gx-Man/DiMan, which showed non- (*n* = 1) and negative- (*n* < 1) cooperativity against DC-SIGN- and DC-SIGNR- mediated viral infection, respectively [14,21]. The different cooperativity of Gx-Man/DiMan blocking DC-SIGN- from DC-SIGNR- mediated viral infection was assigned to their distinct binding modes, with DC-SIGN forming 1:1 tetravalent binding to a single Gx-Man/DiMan particle, while DC-SIGN crosslinking with multiple Gx-Man/DiMan particles. In that case, binding of a single Gx-Man/DiMan particle onto a cell surface DC-SIGN receptor could completely block it from further binding to EBOVpps to initiate infection. In contrast, binding of Gx-Man/DiMan to multiple cell surface DC-SIGNR receptors via crosslinking would be difficult to completely block DC-SIGNR’s CRDs from further binding with EBOVpps to facilitate infection [14,21].

The inhibition of DC-SIGNR-mediated EBOVpp entry by Gx-ManA was found to greatly improved with the increasing GNP scaffold size, as reflected by a greatly reduced *EC*_50_ value (e.g., 18.9 ± 4.9 nM for G5-ManA and 0.29 ± 0.12 nM for G13-ManA), consistent with the trend observed for DC-SIGN-mediated infections. Moreover, Gx-ManAs exhibited negative inhibition cooperativity (*n* < 1), consistent with their Gx-Man/DiMan counterpart [14,21]. Interestingly, while Gx-ManA exhibited strong dose-dependent inhibition of DC-SIGN/R mediated EBOVpp infection as expected, they also dose-dependently inhibited (to a lesser extend) the control particle bearing the VSV-G which does not rely on DC-SIGN/R for cell entry. Thus, Gx-ManA appeared to be a less specific virus inhibitor than Gx-Man/DiMan. This is understandable because Gx-ManA particles are negatively charged, due to deprotonation of the ManA CO_2_H groups at physiological pH, allowing them to electrostatically interact with other non-target (positively charged) viral proteins to partially inhibit viral entry. In contrast, Gx-Man/DiMan are charge neutral and only bind to target lectins via specific MLGIs. Therefore, negatively charged Gx-ManA particles may exhibit broader viral inhibition properties but with lower specificity than its neutral Gx-Man/DiMan counterparts. Overall, our results indicate that increasing the GNP scaffold size can significantly enhance Gx-ManA’s ability to block DC-SIGN/R-promoted viral infection.

Interestingly, whilst the *K*_d_s and *EC*_50_ values of Gx-ManA probes derived from fluorescence quenching and viral inhibition assays do not match directly, likely due to different assay environments (i.e., solution-based measurements vs. membrane-associated interactions), a clear positive correlation between the *K*_d_ and *EC*_50_ values is evident. For example, Gx-ManAs showing stronger affinity (lower *K*_d_) consistently gave higher antiviral potency (lower *EC*_50_), suggesting that our GNP-based fluorescence quenching-derived affinity measurement can serve as a reliable, rapid method for predicting the relative potencies of GNP-glycan based viral entry inhibitors. 

## 4. Conclusions

In summary, we investigated the impact of GNP scaffold size on their MLGI affinity and antiviral properties for mannuronic acid coated GNPs (Gx-ManA). We quantitatively assessed the binding affinities between Gx-ManA and two lectin-based virus receptors, DC-SIGN and DC-SIGNR, using our established NSET-based fluorescence quenching assay. We revealed that increasing the GNP scaffold size from 5 to 13 nm substantially enhanced their MLGI affinities for both DC-SIGN/R, consistent with our earlier results obtained with GNPs coated with lectins’ natural glycan ligands. Particularly, G13-ManA displayed ∼6 times lower *K*_d_s in binding to both DC-SIGN/R compared to G5-ManA. Consistent with these results, the GNP scaffold size was found to play a critical role in determining Gx-ManA’s antiviral property. A modest increase of GNP size from 5 to 13 nm in Gx-ManA leads to a markedly enhanced antiviral potency (e.g., ~19 and ~64 folds against DC-SIGN- and DC-SIGNR- mediated EBOVpp infection, respectively). Compared to Gx-Man/DiMan, Gx-ManA exhibited broader antiviral activities but with lower specificity, possibly due to its negatively charged nature. A clear positive correlation between Gx-ManA’ MLGI affinity with DC-SIGN/R (*K*_d_, measured in solution by GNP-based fluorescence quenching) and their antiviral potency (*EC*_50_, measured on cell membranes), suggesting that GNP-based fluorescence quenching affinity measurement can serve as a reliable, rapid method for predicting the antiviral potentials of GNP-glycans. Overall, our results have revealed the critical importance of GNP scaffold size in controlling GNP-glycan MLGI affinities and antiviral efficacies. These findings can provide useful insights into the design of next-generation glyconanoparticle therapeutics to potently and specifically block lectin-mediated viral infections.

## Data Availability

Data are contained within the article and Appendix A.

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
