# Peer review of "Polyvalent Mannuronic Acid-Coated Gold Nanoparticles for Probing Multivalent Lectin–Glycan Interaction and Blocking Virus Infection"

_viruses, 2025, doi:10.3390/v17081066_

Round 1
Reviewer 1 Report
Comments and Suggestions for Authors
Major comments
This manuscript describes that using polyvalent carboxyl mannose-coated gold nanoparticles (GNP-Man-A) for blacking Ebola virus infection. This approach is based on the lectin-glycan binding interaction to create GNP-Man-A particles for blocking viral infection. Because the GNP-Man-A can interfere viral binding to the lectin-based attachment factors (DC-SIGN and DC-SIGNR) for viral infection. Interestingly, they found increasing the GNP size can significantly enhance the binding to DC-SING/R and also enhance the antiviral potency. Based on testing from pseudotyped Ebola viruses, they have achieved ~1uM EC50 value of neutralization using G13 (13nM) GNPs. It is suggested that GNP-Man-A approach may have the potential for therapeutic applications against Ebola virus infection.
The manuscript is well written, and data is also well presented. GNP synthesis procedures and QC data have been well documented in the supplemental materials.
Minor comments
- How were these two GNP sizes (G5 and G13 ) chosen for this study?
- Line 87, the term “viral lectins” should be changed, such as lectin-based viral attachment factors.
- Any toxicity of these GNPs?
Author Response
Reviewer 1
This manuscript describes that using polyvalent carboxyl mannose-coated gold nanoparticles (GNP-Man-A) for blacking Ebola virus infection. This approach is based on the lectin-glycan binding interaction to create GNP-Man-A particles for blocking viral infection. Because the GNP-Man-A can interfere viral binding to the lectin-based attachment factors (DC-SIGN and DC-SIGNR) for viral infection. Interestingly, they found increasing the GNP size can significantly enhance the binding to DC-SING/R and also enhance the antiviral potency. Based on testing from pseudotyped Ebola viruses, they have achieved ~1uM EC50 value of neutralization using G13 (13nM) GNPs. It is suggested that GNP-Man-A approach may have the potential for therapeutic applications against Ebola virus infection.
The manuscript is well written, and data is also well presented. GNP synthesis procedures and QC data have been well documented in the supplemental materials.
Reply: We thank this reviewer for their recognition and strong supportive comments.
Minor comments
Comment 1: How were these two GNP sizes (G5 and G13) chosen for this study?
Reply: We thank the Reviewer for this comment. We have now clarified our rationale for selecting these two different size nanoparticles in the revised manuscript. See revised manu, lines 385-389.
Interestingly, the LA-EG2-ManA ligand-coated G5 has a similar Dh size to a gp160 trimer, the HIV surface densely glycosylated spike protein, which is responsible for mediating HIV-DC-SIGN interaction and viral infection.37, 38 Thus it was selected to mimic gp160-DC-SIGN interactions. The larger G13-ManA was employed to investigate the impact of GNP scaffold size on their MLGI properties with DC-SIGN/R.
Comment 2: Line 87, the term “viral lectins” should be changed, such as lectin-based viral attachment factors.
Reply: We thank this Reviewer for pointing out this issue. We have now changed the term “viral lectins” to “lectin-based virus receptors”. See revised manu, lines 87 and 623, highlighted sections.
Comment 3: Any toxicity of these GNPs?
Reply: We thank this Reviewer for highlighting this important aspect. While we acknowledge that toxicity of nanomaterials is key factor to consider for potential real-world applications, the primary focus of this study is to investigate the biophysical interactions between ManA-coated GNPs and DC-SIGN/R receptors, as well as their properties in blocking viral entry to host cells. While we have not specifically tested the toxicity of G5/G13-ManA conjugates here, we did not observe any detectable cytotoxic effects of G5/G13-ManA toward 293T cells throughput the concentration range used in the antiviral tests. This result also matches well to our expectation that the G5/13-ManA conjugates should pose no significant toxicity because all the components used to make them are biocompatible and of low-/non-toxicity: GNPs are known to have excellent biocompatibility with low-/non-toxicity and do not induce any acute toxicity. Other components, lipoic acid (LA), EGn linkers and ManA used to make up the multifunctional ligands are either essential, natural cellular antioxidant components (LA) or known to be biocompatible and non-cytotoxic (EGn, ManA). Therefore, we do not believe toxicity will be a significant issue for the GNPs used here. Despite these, we acknowledge that this is importance for potential translation applications which will be addressed in a follow-on study in the future.
Reviewer 2 Report
Comments and Suggestions for Authors
- For a 1:1 binding interaction without cooperativity, accurate determination of Kd requires a titration range of at least five orders of magnitude to capture the full sigmoidal response. In Fig. 4, the lectin concentrations (0–80 nM) is insufficient for obtaining accurate Kd.
- The 1:1 binding interaction model seems to contradict the following statement (line 577-578): “This may indicate that G13-ManA can simultaneously bind to all four CRDs in one DC-SIGN receptor.” If the binding is multivalent, can the 1:1 binding interaction model be used in the calculations?
- Are the Kd and EC50 values (nM) the concentrations of the GNP or the Man ligand on the nanoparticle? This could affect the conclusions on the size-dependent binding affinity and inhibition activity.
- The statement “reduced surface curvature of the larger GNP scaffold increases glycan accessibility” (lines 502–503) appears counterintuitive. Typically, higher surface curvature as found in smaller nanoparticles reduces steric hindrance of surface-bound ligands, thereby enhancing their accessibility for receptor binding. It would be helpful to compare the current work with literature and include discussions on what may contribute to the observed results.
- The authors found that “increasing the size of GNP significantly enhances their MLGI affinity”. Again, this contrasts with vast literature showing lower protein binding with increasing particle size.
- Please clarify the rationale of including a carboxyl on the Man ligand, whether the carboxyl group plays a structural or functional role in receptor binding.
- Proper nomenclature for α-mannose and α-manno-α-1,2-biose should be used. For example, α-mannose should be D-mannose.
Author Response
Comment 1: For a 1:1 binding interaction without cooperativity, accurate determination of Kd requires a titration range of at least five orders of magnitude to capture the full sigmoidal response. In Fig. 4, the lectin concentrations (0–80 nM) is insufficient for obtaining accurate Kd.
Reply: We disagree with this comment. The principles of our GNP fluorescence quenching method for Kd determination are given in the main text, see revised manu, lines 450-479.
Theoretically, it is possible to obtain accurate Kd for a reversible 1: 1 binding interaction at a single concentration, if we can accurately measure the percentage of bound species at equilibrium.
For a reversible 1:1 binding interaction,
A + B ↔ AB (1)
Kd = [A][B]/[AB] (2)
Where [A], [B] and [AB] are the equilibrium concentrations of A, B, and AB complex, respectively.
Starting with the same initial concentration of C0 for both A and B, assuming x% of them are bound to form the AB complex, then the equilibrium concentration of the AB complex, [AB] = C0 x%, while the free A and B concentrations at equilibrium, [A] = [B] = C0 (100% - x%).
Substituting these data to equation (2) gives,
Kd = [A][B]/[AB] = {C0 (100% - x%)}2/(C0 x%) = C0 (100% - x%)2/x% (3)
Therefore, if x% can be measured accurately at a single starting concentration C0, then Kd can be accurately determined from this fundamental chemical binding equilibrium equation (3).
Note, this equation is derived from the standard 1: 1 reversible binding interactions between A and B without proximation. In this study, where A = GNP-ManA, B = DC-SIGN, and AB = GNP-ManA-DC-SIGN complex.
Given GNPs can efficiently quench fluorophores at proximity, up to 99.97% (see Dubertret et al., Nat Biotechnol, 2001, 19, 365), it is safe to assume that any DC-SIGN molecules bound to GNPs are fully (~100%) quenched, and thus quenching efficiency (QE%) is an accurate measure of the percentage of DC-SIGN molecules bound to GNPs.
In this way, accurate Kds can be measured at a single starting concentration (C0) for both binding partners by measuring the percentage of bound species at that concentration. In this study, we have measured QE% at several concentration points and performed a linear fit to improve the accuracy of QE% measurement.
The reliability of our GNP fluorescence quenching based Kd measurement method has already been proven in several of our earlier publications, (see Budhadev et al., J. Am. Chem. Soc., 2020, 142, 18022; Ning et al., JACS Au, 2024, 4, 3295; Basaran et al., Nanoscale, 2024, 16, 13962).
Although we noticed several other MLGI Kd measurement method reported in literature. For example, Yan’s group has used free sugars (with known monovalent Kdmono with target lectin) to compete with GNP-glycans binding to fluorophore-labelled lectin, leading to disassembly and release of labelled lectins from the bound lectin-GNP-glycan complexes in a dose dependent manner. By fitting the released lectin fluorescence against free sugar concentrations (over 4-5 orders of magnitude) using a competition equation, the Kd of MLGIs between GNP-glycans and target lectins (see S. H. Liyanage and M. Yan, Chem. Commun., 2020, 56, 13491). However, that is an indirect and completely different Kd measurement method from ours described here.
Comment 2: The 1:1 binding interaction model seems to contradict the following statement (line 577-578): “This may indicate that G13-ManA can simultaneously bind to all four CRDs in one DC-SIGN receptor.” If the binding is multivalent, can the 1:1 binding interaction model be used in the calculations?
Reply: We thank this Reviewer for this comment. Herein, we’d like to clarify that the 1: 1 binding model here refers one Gx-ManA particle binding to one tetrameric DC-SIGN molecule, not one ManA residue binding to one CRD. Here, each Gx-ManA contains ~530 (for G5) or ~1800 (for G13) ManA residuals on its surface, while each DC-SIGN molecule contains 4 CRDs with all 4 binding sites pointing upwardly away from the coiled-coil neck as shown schematically in Figure 1B. Given the high glycan polyvalency and a large surface area, each GNP-glycan can bind to > 10 DC-SIGN molecules before reaching saturation. Moreover, DC-SIGN forms a stable tetramer and binds tetravalently (with all 4 CRDs) to a single GNP-glycan, where binding of multiple DC-SIGNs to one GNP-glycan forms a monolayer DC-SIGN coating surrounding a single GNP-glycan (see Budhadev et al., J. Am. Chem. Soc., 2020, 142, 18022; Ning et al., JACS Au, 2024, 4, 3295; Basaran et al., Nanoscale, 2024, 16, 13962). Based on these evidences, it is safe to treat one Gx-ManA binding to one DC-SIGN molecule as the standard 1 : 1 binding interaction.
Changes: to reflect this comment and avoid misunderstanding, we have now clarified our 1:1 binding model as single Gx-ManA binding to single DC-SIGN molecules in the revised manu. See highlighted sections in line 458 and line 591-593.
Comment 3: Are the Kd and EC50 values (nM) the concentrations of the GNP or the Man ligand on the nanoparticle? This could affect the conclusions on the size-dependent binding affinity and inhibition activity.
Reply: We thank this Reviewer for the feedback. The reported Kd and EC50 values are both referred as the concentrations of GNP, not the concentration of ManA. The corresponding ManA concentration is ~530 (for G5) or ~1800 (for G13) fold that of Kd or EC50 values because each G5 or G13 contains ~530 or ~1800 copies of LA-EG2-ManA ligand.
Changes: we have now clarified that both the Kd and EC50 values refers Gx-ManA particle concentration. See revised manu, highlighted sections, lines 485-486 and 571-572.
Comment 4: The statement “reduced surface curvature of the larger GNP scaffold increases glycan accessibility” (lines 502–503) appears counterintuitive. Typically, higher surface curvature as found in smaller nanoparticles reduces steric hindrance of surface-bound ligands, thereby enhancing their accessibility for receptor binding. It would be helpful to compare the current work with literature and include discussions on what may contribute to the observed results.
Reply: We thank this Reviewer for this comment. We agree that a higher surface curvature afforded by a smaller nanoparticle can reduce steric crowding of ligands (at the same surface density), which can potentially enhance ligand accessibility to lectin binding. However, the effect here will depend on the binding mode of multimeric lectins. While the effect may be true for those with crosslinking binding mode (i.e., one lectin crosslinks with multiple GNP-glycan particles, such as ConA and a few other lectins reported in literature, see Liyanage & Yan, Chem. Commun., 2020, 56, 13491), it is not true for those forming simultaneous 1: 1 binding (one lectin to one GNP-glycan) with a single GNP-glycan, such as DC-SIGN here. Because all 4 binding sites in one DC-SIGN molecule are roughly on the same plane and point upwardly from the coiled-coil neck, DC-SIGN prefers to bind tetravalently, with all 4 CRDs, to glycans on a single GNP. In this regard, a flatter display of glycans afforded by the larger G13 scaffold over the smaller G5 presents a better topological match for DC-SIGN to form 1: 1 tetravalent binding (as shown schematically below in Figure S1). This binding mode is more thermodynamically favourable than crosslinking because not only can it maximize the MLGI enthalpy (engaging with all 4 binding sites) but also minimize the unfavourable entropic penalty (by forming small 1:1 lectin-GNP-glycan complexes rather than large scale lectin-GNP-glycan assemblies).
Figure R1: Schematic show of DC-SIGN binding to G5-ManA (A) and G13-ManA (B) where blue dots represent ManA residues. The small non-deformable G5 core creates a highly curved ManA display, which is difficult to bind simultaneously to DC-SIGN’s all 4 glycan binding sites to form 1: 1 tetravalent binding. In contrast, G13 creates a much flatter display of ManA surface, allowing it to readily bind to all 4 glycan binding sites on the same DC-SIGN molecule, leading to the formation of strong 1: 1 tetravalent binding with greatly enhanced affinity.
Changes: we have now added a few sentences to discuss such differences. See revised manu, lines 507-517, highlighted section. We also included two new relevant references, Wang et al., Anal. Chem., 2010, 82, 9082. Liyanage & Yan, Chem. Commun., 2020, 56, 13491.
Comment 5: The authors found that “increasing the size of GNP significantly enhances their MLGI affinity”. Again, this contrasts with vast literature showing lower protein binding with increasing particle size.
Reply: We thank this Reviewer for this comment. The dependence of MLGI affinity on GNP scaffold size is not simple and is strongly affected by factors such as linker length, flexibility, water solubility, and more critically, binding mode with the last factor being rarely mentioned. Because previous literature studies have been mainly based on lectins with crosslinking binding mode, e.g., ConA, PA-IL, peanut agglutinin (PNA), Cyanovirin-N (CVN), etc.. In those cases, their MLGI affinities with GNP-glycans are found to strongly depend on linker length, hydrophilicity and flexibility, with a flexible, hydrophilic linker generally enhancing their MLGI affinity (see review, Liyanage & Yan, Chem. Commun., 2020, 56, 13491). The dependence of MLGI affinity on GNP scaffold size is less clear cut, where examples of both increasing (see Lin et al., Chem. Commun., 2003, 2920; Chien et al., ChemBioChem, 2008, 9, 1100) and decreasing (see Liyanage & Yan, Chem. Commun., 2020, 56, 13491) MLGI affinity with the increasing GNP size have been reported. For lectins showing 1:1 multivalent binding with one GNP-glycans, such as DC-SIGN here, we have shown recently that increasing GNP scaffold size can significantly enhance their DC-SIGN MLGI affinity with both natural DiMan ligand (see Basaran et al., Nanoscale, 2024, 16, 13962) and a synthetic glycomimetic psDiMan ligand (see Ning et al., JACS Au, 2024, 4, 3295). The same conclusion has been further confirmed here with the ManA ligand.
Changes: We have clarified that the enhanced MLGI affinity with the increasing GNP scaffold size here applies for DC-SIGN/R. We did not state that this effect is applicable to other lectins. This result is consistent to our earlier results obtained for GNPs coated with both DC-SIGN’s natural glycan, Gx-DiMan, and a synthetic glycomimetic ligand, Gx-psDiMan. See revised manu, highlighted section in lines 518-520.
A new relevant ref (Ning et al., JACS Au, 2024, 4, 3295) has been cited.
Comment 6: Please clarify the rationale of including a carboxyl on the Man ligand, whether the carboxyl group plays a structural or functional role in receptor binding.
Reply: We thank the Reviewer for raising this point. The introduction of a carboxyl group at the C6 position of mannose (to yield ManA) was intended to probe how such a structural change affects their DC-SIGN/R MLGI affinity and antiviral efficacy. The binding of mannose to DC-SIGN/R is mainly initiated by coordination of its 3,4-hydroxyl groups to the Ca2+ ion in CRD binding site (see Feinberg et al., Science 2001, 294, 2163) by converting the C6 -CH2OH group in mannose into a -CO2H group in ManA, we have anticipated that it might provide additional Ca2+ coordination, hydrogen bonding and/or electrostatic interactions to enhance its CRD binding affinity. We have found that while G5-ManA exhibits a slightly weaker MLGI affinity with DC-SIGN than G5-Man does (Kd: 61 ± 3 nM vs. 33 ± 2 nM), its MLGI affinity with DC-SIGNR is notably enhanced than G5-Man (Kd, 48 ± 4 nM vs. 214 ± 68 nM), making it more selective toward DC-SIGNR over DC-SIGN.
Changes: we have now added the rationale of our choice of ManA here. See revised manu, lines 133-135, highlighted section. A relevant new ref. (Feinberg et al., Science 2001, 294, 2163) has also been added.
Comment 7: Proper nomenclature for α-mannose and α-manno-α-1,2-biose should be used. For example, α-mannose should be D-mannose.
We thank this Reviewer for this comment and suggestion regarding carbohydrate nomenclature. We fully agree that precision in chemical terminology is essential for clarity, consistency, and standardization in scientific communication. In our manuscript, the terms “α-mannose” and “α-manno-α-1,2-biose” were intentionally used to emphasize the anomeric configuration and glycosidic linkage, which are known to be critical for lectin–glycan recognition. Such names are widely employed in glycobiology literatures to highlight the biological relevance of carbohydrate structures, and particularly when discussing lectin binding preferences and molecular recognition mechanisms. Because lectins exhibit high selectivity not only for glycan identity but also for their orientation and glycosidic linkages, the use of such structurally descriptive names underscore these specificities. We believe that it is appropriate to retain these terms in biologically contextual discussions, which can improve interpretability and clarity, especially for readers with a biochemical and/or glycoscience background. It also provides a meaningful balance between biochemical clarity and structural accuracy.

Round 2
Reviewer 2 Report
Comments and Suggestions for Authors
The revision is largely fine. One question regarding Fig. 4, where the lectin concentration was in the range of 0–80 nM. What is the rationale of selecting this concentration range? What would be the result, (QE × C) versus C, in concentrations higher than 80 nM? Will one obtain a linear fit? Please include a comment/discussion on this.
Author Response
Comment: The revision is largely fine. One question regarding Fig. 4, where the lectin concentration was in the range of 0–80 nM. What is the rationale of selecting this concentration range? What would be the result, (QE × C) versus C, in concentrations higher than 80 nM? Will one obtain a linear fit? Please include a comment/discussion on this.
Reply: Thank you for highlighting this. We have, accordingly, revised the manu and corrected all ref format. Changes are highlighted in lines 434-445 and 487-491.
